# Atomic structure of a nudivirus occlusion body protein determined from a 70-year-old crystal sample

Jeremy R. Keown [1] ✉, Adam D. Crawshaw [2], Jose Trincao[2], Loïc Carrique[1], Richard J. Gildea [2], Sam Horrell [2], Anna J. Warren [2], Danny Axford [2], Robin Owen [2], Gwyndaf Evans [2,3], Annie Bézier [4], Peter Metcalf [5] & Jonathan M. Grimes [1] ✉

Infectious protein crystals are an essential part of the viral lifecycle for double-stranded DNA *Baculoviridae* and double-stranded RNA cypoviruses. These viral protein crystals, termed occlusion bodies or polyhedra, are dense protein assemblies that form a crystalline array, encasing newly formed virions. Here, using X-ray crystallography we determine the structure of a polyhedrin from *Nudiviridae*. This double-stranded DNA virus family is a sister-group to the baculoviruses, whose members were thought to lack occlusion bodies. The 70-year-old sample contains a well-ordered lattice formed by a predominantly α-helical building block that assembles into a dense, highly interconnected protein crystal. The lattice is maintained by extensive hydrophobic and electrostatic interactions, disulfide bonds, and domain switching. The resulting lattice is resistant to most environmental stresses. Comparison of this structure to baculovirus or cypovirus polyhedra shows a distinct protein structure, crystal space group, and unit cell dimensions, however, all polyhedra utilise common principles of occlusion body assembly.

Viral occlusion bodies (OBs), also called polyhedra, are stable protein crystals that encapsulate newly assembled virions within virus-infected cells[1]. They most commonly occur in larvae infected with insect viruses from the *Baculoviridae, Reoviridae, and Poxviridae* families and function as the environmentally persistent infectious form of these viruses. These OBs were first described in the early 20th century where they were depicted to be highly reflective and have a polyhedron geometry (formed by flat polygons with straight edges and vertices)[2]. As technologies became available, electron microscopy was used to demonstrate that these OBs contained virus particles. Isolation of significant amounts of OBs allowed early proteomics approaches to identify that they were predominantly formed from a single protein which was subsequently partially sequenced[3–6]. Genome sequencing was then used to identify full-length polyhedrin protein sequences for recombinant expression.

Cells infected with these insect viruses produce polyhedra that range in size from 0.2 to 10 microns and consist primarily of a single 25–33 kDa polypeptide polyhedrin[1,4]. The crystalline OB protein matrix protects the embedded virions from environmental conditions including dehydration, freeze-thaw, and high temperatures[4,7]. In addition, polyhedra resist proteolytic digestion by enzymes and chemical disruption by chaotropic agents, concentrated acids, detergents, and organic solvents[7]. Despite their resistance to many environmental conditions, they readily dissolve at pH >10.5[7–9]. This dissolution under alkali pH allows virion release and infection in the alkaline midgut of insect larvae which is one of the main routes of viral infection[2,4,7–9].

[1]Division of Structural Biology, Wellcome Centre for Human Genetics, University of Oxford, Oxford, UK. [2]Diamond Light Source Ltd, Harwell Science & Innovation Campus, Didcot, UK. [3]Rosalind Franklin Institute, Harwell Campus, Didcot, UK. [4]Institut de Recherche sur la Biologie de l'Insecte (IRBI), UMR7261 CNRS-Université de Tours, Tours, France. [5]School of Biological Sciences, University of Auckland, Private Bag 92019, Auckland, New Zealand. ✉e-mail: Jeremy@strubi.ox.ac.uk; Jonathan@strubi.ox.ac.uk

Native OBs purified from moth larvae infected with cypovirus members of the *Reoviridae* and *Baculoviridae* have been structurally characterised, together with intracellular polyhedrin microcrystals produced by expressing the corresponding recombinant polyhedrin genes in Sf9 insect cells (Reviewed in Chui et al.[1]). The *Baculoviridae* are a large family of circular double-stranded DNA (dsDNA) viruses[10] that form OBs in the nucleus while cypoviruses are segmented linear double-stranded RNA (dsRNA) viruses from the *Reoviridae* family[11] that form cytoplasmic OBs.

Polyhedrin amino acid sequences in general have little sequence conservation and there is no evident homology between baculovirus and cypovirus polyhedrins[12]. Interestingly, the OBs from baculoviruses and cypoviruses have very similar crystal lattices, with cubic space group symmetry I23 and unit cell dimensions between 101 and 106 Å ($a = b = c$)[7–9]. Comparison of structures from the two groups reveals each contains a predominantly β-strand fold with surrounding α-helical extensions, however, differences in the secondary structure topology mean the baculovirus and cypovirus structures are likely unrelated[12]. Although initially considered to be the only viruses using polyhedra, there were some reports of OBs forming in nudivirus-infected cells[13–15].

Nudiviruses are circular dsDNA viruses that infect various insects as well as some aquatic arthropods (recently reviewed in Petersen et al. 2022[16]). Nudiviruses were initially characterised as a sub-group of baculoviruses. They were later reclassified into their own group and given the name nudi, from the latin "*nudus*", due the absence of OBs among species studied at the time. Subsequently, other differences including host range, genome structure, and cytopathology validated this split. However, the initial criteria for the group were invalidated with the observation of OBs for nudivirus species infecting the giant tiger prawn (*Penaeus monodon*) and the marsh crane fly (*Tipula oleracea*), named the *Penaeus monodon* nudivirus (PmNV) and the *Tipula oleracea* nudivirus (ToNV), respectively[13,14]. Sequence analysis identified the putative gene encoding the polyhedrin protein; however, these proteins had little homology either to each other, or to the polyhedrins from baculovirus or cypovirus[14,17].

Nudiviruses are an economically important family of viruses. As biocontrol agents, they have been used to control invasive species. *Oryctes rhinoceros* nudivirus (OrNV) is used to control the coconut palm beetle which infects young plants in parts of Asia[18–20]. In parts of North America, the marsh crane fly, the known target of ToNV, is considered as an invasive pest[21]. Other nudiviruses can also cause extensive damage in crustacean or insect large-scale farming[13,22]. A greater understanding of OBs may allow for the best deployment of nudiviruses as biocontrol agents.

In the present study, we sought to understand whether the structural features and lattice constraints observed are conserved in examples of OBs identified in nudiviruses. To do this we determined the structure of the ToNV polyhedrin from native 70-year-old polyhedra and recombinantly expressed and assembled protein. In an interesting example of convergent evolution, our data show a conservation of polyhedra design principles using a unique protein building block.

## Results

### Determining the structure of the ToNV polyhedrin
Previous characterisation of the native OB showed particles with sizes of 2–5 microns with an irregular morphology that still contained infectious viral particles[14]. Initial X-ray experiments yielded high-quality diffraction patterns that could be readily indexed and gave good data collection statistics to a high resolution. To generate experimental phases, the wild-type ToNV polyhedrin protein was cloned into an insect cell expression vector before three hydrophobic residues (F104/L105/L137) were mutated to methionine. The native sequence contains only an N-terminal methionine. The structure was determined using single wavelength anomalous diffraction (SAD) data

measured using the nano/microfocus X-ray diffraction beamline VMXm at Diamond Light Source that employs a low-background, in-vacuum cryogenic sample environment. To prepare samples for the VMXm beamline, a slurry of microcrystals was first applied to a single cryoEM grid that was then vitrified (Fig. 1a). The crystals on a single grid were then used for diffraction experiments, enabling the efficient collection of data from many protein crystals.

X-ray diffraction data for both recombinant selenomethionine derivatised and native polyhedra were measured from protein crystals mounted on cryoelectron microscopy grids (Fig. 1a). SAD data were collected from selenomethionine crystals, with dimensions between 1–5 microns using a beam size of 3 by 4 (H × V) µm. Due to the small size of the crystals, small wedges of data were collected per crystal and then merged to form a final dataset using data from 55 crystals. The model for the native polyhedra was more complete and determined to a higher resolution and will be the dataset described unless otherwise stated. In our structure we observe a single copy of the polypeptide in the asymmetric unit, with a solvent content of ~22% (Matthew's coefficient of 1.59 Å³/Da) placing this in the bottom ~0.1% of structures by decreasing crystal solvent content in the Protein Data Bank. The crystals belong to space group P3$_2$21 with unit cell dimensions of a = b = 53.7 Å, c = 105.6 Å (Supplementary Table. 1).

Attempts to phase the datasets with molecular replacement using previously determined polyhedra structures, in silico structures (trRosetta and AlphaFold), or secondary structure fragments were trialled but ultimately unsuccessful (Supplementary Fig. 1a)[23–25]. In silico models of oligomeric ToNV polyhedrin were generated, however these were also unsuccessful in molecular replacement.

### Structure of the ToNV polyhedrin
The crystalline polyhedra is formed by repeating units of the ~27.5 kDa ToNV polyhedrin protein, which is predominantly helical, containing ten helices and a short two-stranded antiparallel beta sheet (Fig. 1b, Supplementary Fig. 2a). The polypeptide was resolved from residues 4–237 with only the internal loop 171–174 and short regions of each terminus insufficiently ordered to be modelled in the electron density map. The N-terminal 38 residues comprising helices-α$_{1-3}$ and an extended linker are separate from the body of the protein, forming contacts with adjacent molecules. The body of the polyhedrin molecule is formed by seven helices-α$_{4-10}$. Secondary structure homology searches using the FoldSeek, Dali, or PDBeFold servers against the experimental structures in the Protein Data Bank or predicted models in the AlphaFold Protein Structure Database did not reveal any structures homologous to ToNV polyhedrin[26–28]. The antiparallel strands β$_{1-2}$ and helices-α$_{4-9}$ pack against one face of the approximately 50 residue C-terminal helix-α$_{10}$. A single calcium ion was observed at the loop between helix-α$_8$ and helix-α$_9$ and is coordinated by the backbone carboxyl groups of residues Asn148 and Asp140, the carboxyl sidechains of Asp140 and Asn143, the amine group from Asn143, and two water molecules. The pentagonal bipyramidal coordination and well-refined atomic position support the ion assignment as calcium. Interestingly, we observe an uncommon cis peptide bond between residues Gly165 and Ile166 prior to an inter-subunit disulfide bond and the disordered loop 171–174 (Supplementary Fig. 2b). To ease description of the crystal lattice and comparison to other OB we will consider a dimer as the minimal repeating unit of the crystal.

This dimer is symmetric, with the interface formed by residues 206–236 from helix-α$_{10}$, residues 114–135 from helix-α$_8$, and the loop between the two strands of β$_1$ and β$_2$ (Fig. 1c, Supplementary Fig. 3a). This interface buries a surface area of 1230 Å² from each molecule and is maintained by extensive hydrophobic and electrostatic interactions. A salt-bridge is formed between Asp135 and Arg211 and hydrogen bonds between Asn132-Asn218, Asp121-Thr228, Lys227-Ser224, and Tyr216-Thr221 (Supplementary Fig. 3b). A hydrophobic interface of

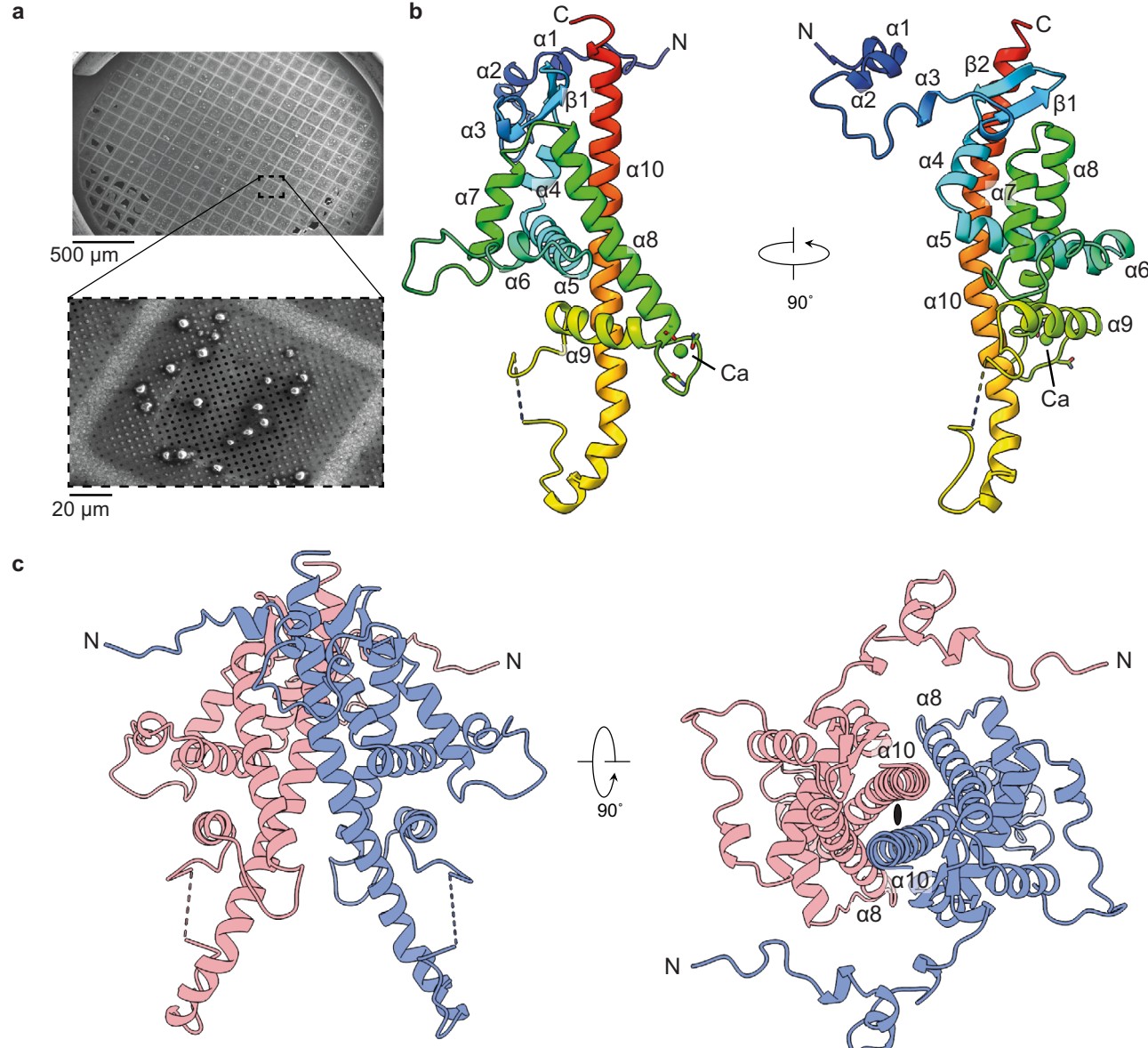

**Fig. 1 | Structure of the ToNV polyhedrin. a** Scanning electron micrograph of native ToNV occlusion bodies prepared for diffraction experiments. Images have been cropped but are otherwise unedited. **b** A single polyhedrin molecule coloured from the N-terminus (blue) to the C-terminus (red) and annotated with secondary structure features. A calcium ion is shown (green). The dashed line shows the missing loop 171–174. **c** The dimeric unit (red and blue chains) of the OB lattice are shown in two orientations. The two-fold symmetric axis is indicated by the black ellipse.

twenty amino acid residues provides a complementary surface against which the second molecule of the dimer packs. Interface analysis using the PISA server (https://www.ebi.ac.uk/pdbe/pisa/) shows this is the most stable interface in the lattice[29].

**Building the crystalline lattice of the polyhedra**
The ToNV polyhedra lattice can be considered as a stack of 2D crystalline sheets. We will first consider the interactions maintaining the 2D lattice before considering the interactions between the sheets. Each polyhedrin molecule interacts with eight adjacent molecules in addition to the molecule with which it forms a dimer (Fig. 2a). For ease of discussion, we number the protomer which forms the repeating dimer (I), and the other adjacent protomers numbers (II-IX) (Fig. 2b). Each protomer forms four disulfide bonds, 34 hydrogen bonds, and 30 salt-bridges with adjacent protomers. These interactions bury approximately 48% or 7518 Å² of the surface area of the protomer.

The N-terminus of the polyhedrin molecule is in an extended conformation, reaching across protomer I to contact protomer II. This interaction is stabilised by disulfide bond formation between Cys5 and Cys167 of protomer II (Fig. 2c). A second disulfide bond forms between Cys119 of the reference polyhedrin and Cys179 of protomer IV (Fig. 2d). As these are asymmetric disulfide bonds across crystallographic axes there are four cysteines involved in disulfide linkages to each polyhedrin. Of the four remaining cysteines, two, Cys139 and Cys207, are positioned at the crystallographic 2-fold axis. The final two cysteines, Cys240 and Cys241, are not ordered and are the two most C-terminal residues of the polypeptide. The packing of the polyhedrin places these two disordered C-terminal cysteine residues in positions where they would likely be in close proximity to the ordered Cys139/Cys207 of the above polyhedrin (Fig. 2e). Inspection of residues Cys139/Cys207 revealed additional electron density adjacent to the two cysteines (Fig. 2f). Attempts to model the complete C-terminus, to identify the residues involved in disulfide formation were

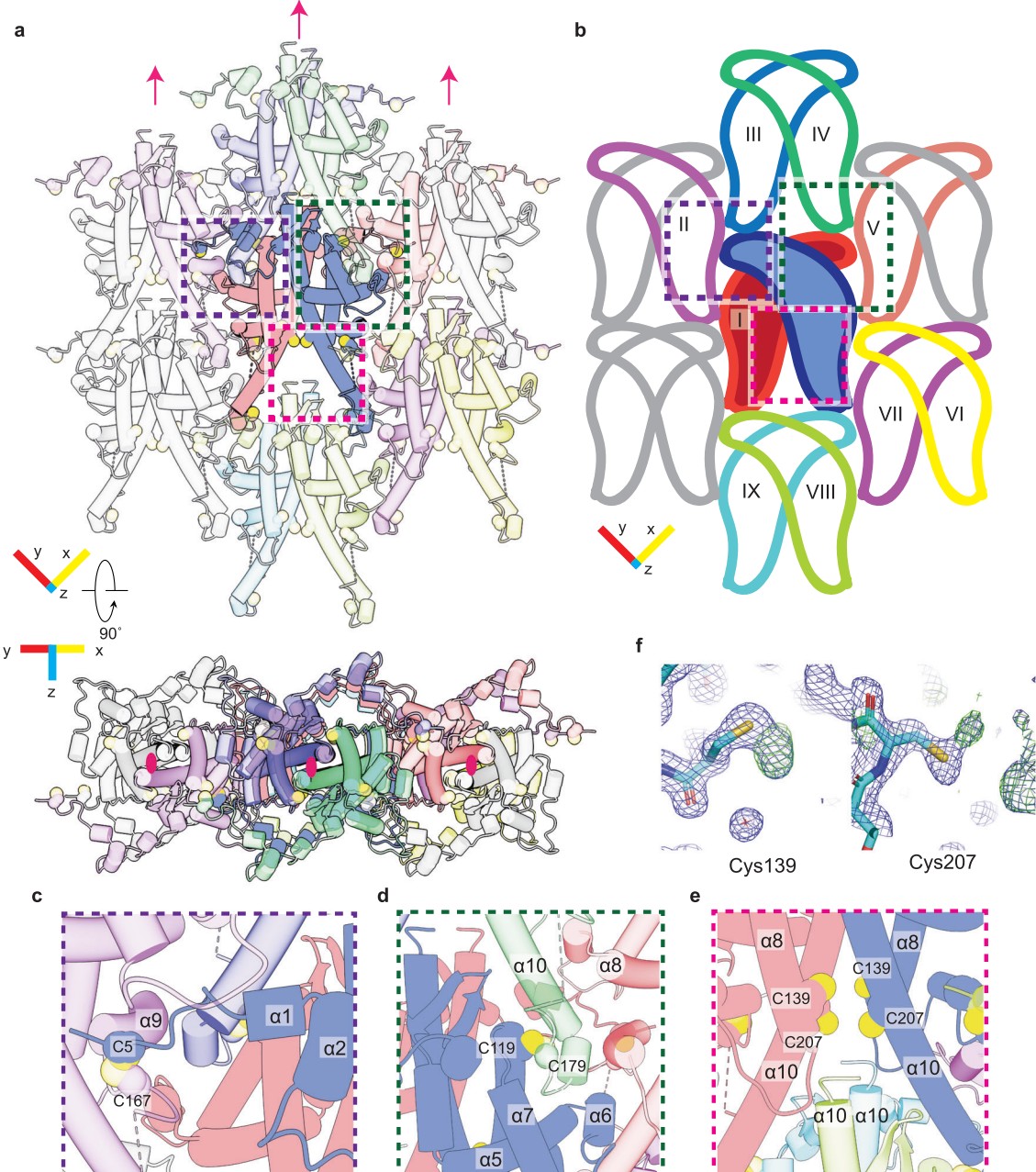

**Fig. 2 | The 2D lattice packs tightly and is maintained through disulfide bond formation. a**, **b** Display of the 2D lattice, the reference polyhedrin (blue) and the polyhedrin with which it makes contact are shown in various colours. The two polyhedrin molecules which form the repeating unit are shown without transparency (**a**) and with fill colour (**b**). Molecules which do not contact the reference molecule are in grey. Pink arrows and ellipses indicate the 2-fold crystallographic axis. **c**, **d** Detailed insets showing the residues which form the asymmetric disulfide bonds. **e** C139/C207 are partially oxidised cysteine residues positioned near to the unordered C-terminus of neighbouring molecules. **f** Additional electron density observed at C139/C207. The $2F_O - Fc$ map at 1σ (blue) and $F_O - Fc$ at 3σ are shown.

unsuccessful. In the crystallographic packing two C-termini will be positioned directly below two sets of residues Cys139/Cys207. Given our inability to trace a unique path for the C-terminal residues and their position on the crystallographic axis, we believe there may be many possible disulfide bonds formed between the C-terminal cysteines and the cysteines of the adjacent molecules. Comparison of the archival sample with recombinant crystals showed the absence of the Cys5-Cys167 disulfide bond, and a disordering of residues near this bond (Supplementary Fig 3c). It may be that this disulfide bond formation is part of the natural maturation of the crystal and the differences seen are due to the protein being produced in a non-native organism. Alternatively, this oxidation may occur as part of the aging

process of the archival sample or an effect of the freeze/thaw cycles during the sample's long life.

With an understanding of how the crystalline lattice arranges in two dimensions, we will now consider the assembly of the lattice in three dimensions. To completely describe the unique interactions between the sheets in the lattice we will discuss three sheets of the crystalline lattice. The central sheet shall be defined as $N_0$ with the sheets immediately in front or behind termed $N_{+1}$ or $N_{-1}$, respectively (Fig. 3a). While the 2D assembly is supported by extensive disulfide bonds, with electrostatic and hydrophobic interactions, the interactions between each sheet are held together by electrostatic and hydrophobic interactions and domain swapping. The sheets of the

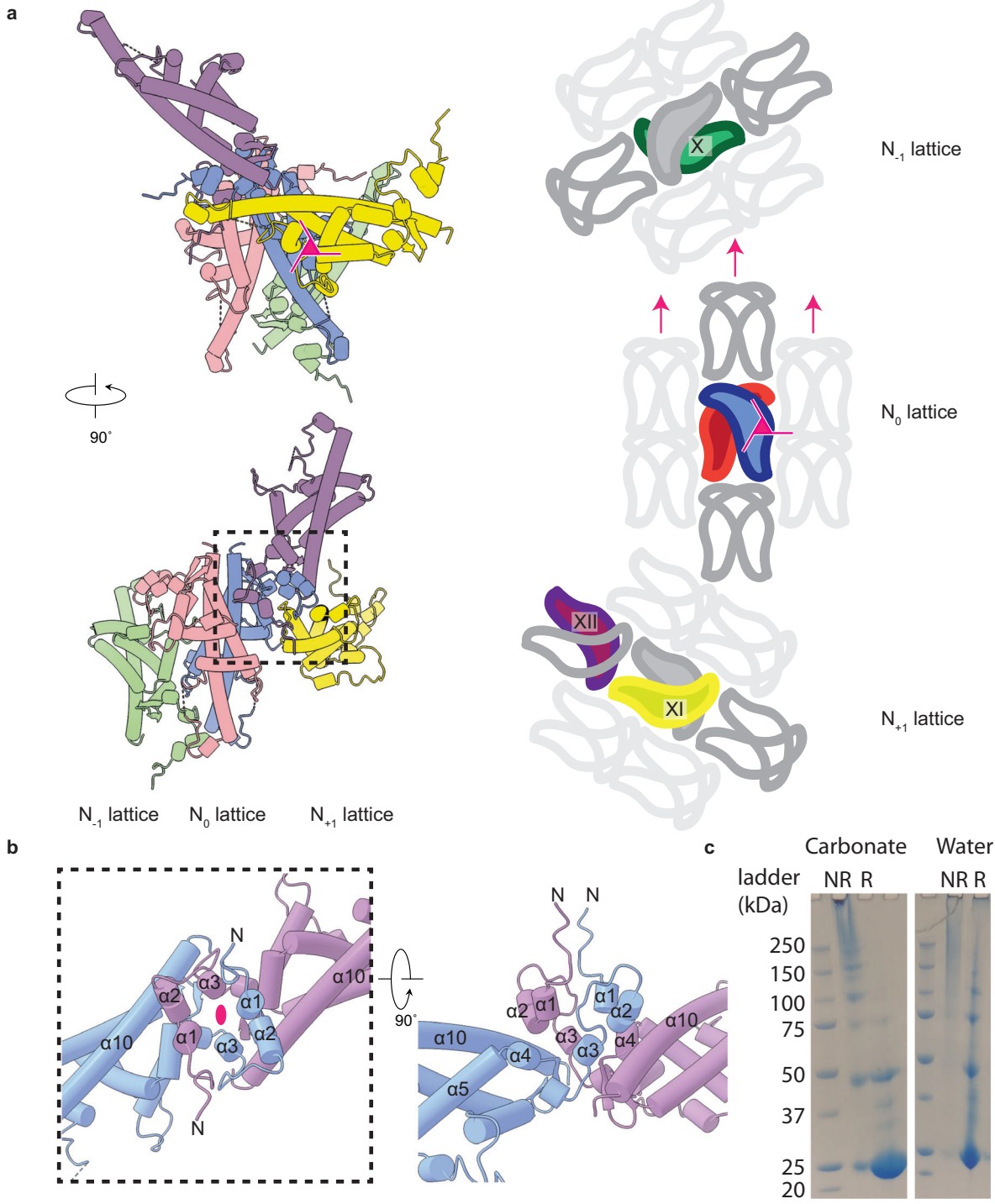

**Fig. 3 | Electrostatic, hydrophobic, and domain swapping maintains interactions between lattice sheets. a** Polyhedrin molecules which form major interfaces between the sheets of the polyhedra. The reference polyhedrin dimer (blue/red), with polyhedrin from the sheets in front (yellow/purple) and behind (green) are shown. A simplified scheme (right side) showing the relative arrangement of polyhedrin around the three-fold crystallographic axis (pink triangle), two-fold axis (pink arrow), and the change in direction of each sequential sheet in the lattice. **b** Detailed view of the domain swapping interactions annotated with secondary structure names. The crystallographic 2-fold axis is shown (pink ellipse). **c** SDS-PAGE of recombinant crystals treated with $NaH_2CO_3$ or water under reducing (R) or non-reducing (NR) conditions. Samples were prepared twice and a representative image shown.

crystal form along the crystallographic 3-fold axis, consequently the $N_{+1}$ lattice is rotated clockwise 120° while the $N_{-1}$ lattice is rotated anticlockwise by the same value (Fig. 3a).

Protomer X, XI, and the reference protomer are related by three-fold rotational crystallographic symmetry. Residues from helix-$\alpha_8$ and the N-terminal region of helix-$\alpha_{10}$ of protomer X from the $N_{-1}$ lattice,

interact with helix-$\alpha_2$, strand-$\beta_1$, and helix-$\alpha_7$ from the reference protomer in the $N_0$ lattice. As this interface is on the crystallographic 3-fold axis these same residues mediate an identical interface between the helix-$\alpha_8$ and the N-terminal region of helix-$\alpha_{10}$ of the reference protomer in lattice $N_0$ and from helix-$\alpha_2$, strand-$\beta_1$, and helix-$\alpha_7$ from protomer XI in lattice $N_{+1}$. The largest single interface in the crystal

lattice is between the reference protomer and protomer XII. This interface is symmetric and is formed through domain swapping of the extended N-terminal 38 residues (Fig. 3b), locking the sheets together. The interface buries 1268 Å² with residues from helices-$\alpha_{1-3}$ swapping over to pack between helices-$\alpha_{1-3}$ and helix-$\alpha_4$/ helix-$\alpha_4$ (Fig. 3b).

The strength of the lattice appears isotropic, with approximately 48% of the protomer interface buried (both hydrophobic, H-bond, and salt bridges) to form the 2D lattice which is further stabilised by at least four disulfide bonds. Between the sheets of the lattice, approximately 44% of the protomer interface is buried and further maintained by extensive domain swapping. In total each polyhedrin buries approximately 92% of the surface area when assembled into the polyhedra, a

similar value to either baculovirus (for example PDB ID: 3JVB) or cypovirus (for example PDB ID: 2OH5) polyhedra at 96% or 98%, respectively. The ToNV polyhedra lattice is not only dense but its arrangement also means that there are no contiguous channels running through the crystal. Comparison of the native (containing virions) or recombinant crystals showed no differences in the crystalline lattice, demonstrating that inclusion of virions does not cause long range disruption of the lattice.

## Lattice stability and dissolution

Analysis of previously described viral polyhedra demonstrated a resistance to harsh biological and non-biological conditions[7–9,13]. Here we sought to assess the stability of the recombinant nudivirus polyhedra by exposing them to environmental conditions and measuring their diffraction (Table 1). The crystals continued to diffract when treated with many conditions including 8 M Urea, 100% ethanol, reducing/oxidising conditions, and harsh physical stress. Crystals treated with 1 M HCl did not dissolve but also did not diffract. Treatment with either 10% w/v Sodium dodecyl sulfate (SDS) or 20 mM NaH$_2$CO$_3$ pH 10.5 was sufficient to completely dissolve the crystals. As these studies were conducted on the recombinant crystals and not the native OB we cannot exclude the possibility the native OB may behave differently. These properties are consistent with those observed for the cypovirus polyhedra[7] and the sensitivity to high pH has been observed for both cypovirus and baculovirus polyhedra[7–9]. The observed dissolution in high pH NaH$_2$CO$_3$ solution was somewhat expected given that this was seen for the other polyhedra and the host for this nudivirus is expected to have a similar midgut pH[30]. This

**Table 1 | Stability of recombinant ToNV crystals**

| Stress type | Condition | Duration (h) | Diffraction |
|---|---|---|---|
| Chaotropic agent | 8 M Urea | 24 | Yes |
| Detergent | 10% w/v SDS | 24 | Dissolved |
| Acid | 1 M HCl | 24 | No |
| Base | 20 mM NaH$_2$CO$_3$ pH 10.5 | 3 or 24 | Dissolved |
| Solvent | 100% Ethanol | 3 or 24 | Yes |
| High molarity | 4 M NaCl | 24 | Yes |
| Oxidising agent | 1% v/v H$_2$O$_2$ | 24 | Yes |
| Reducing agent | 10 mM Dithiothreitol | 24 | Yes |
| Physical stress | 90 °C temperature | 3 | Yes |
|  | Freeze-thaw | – | Yes |

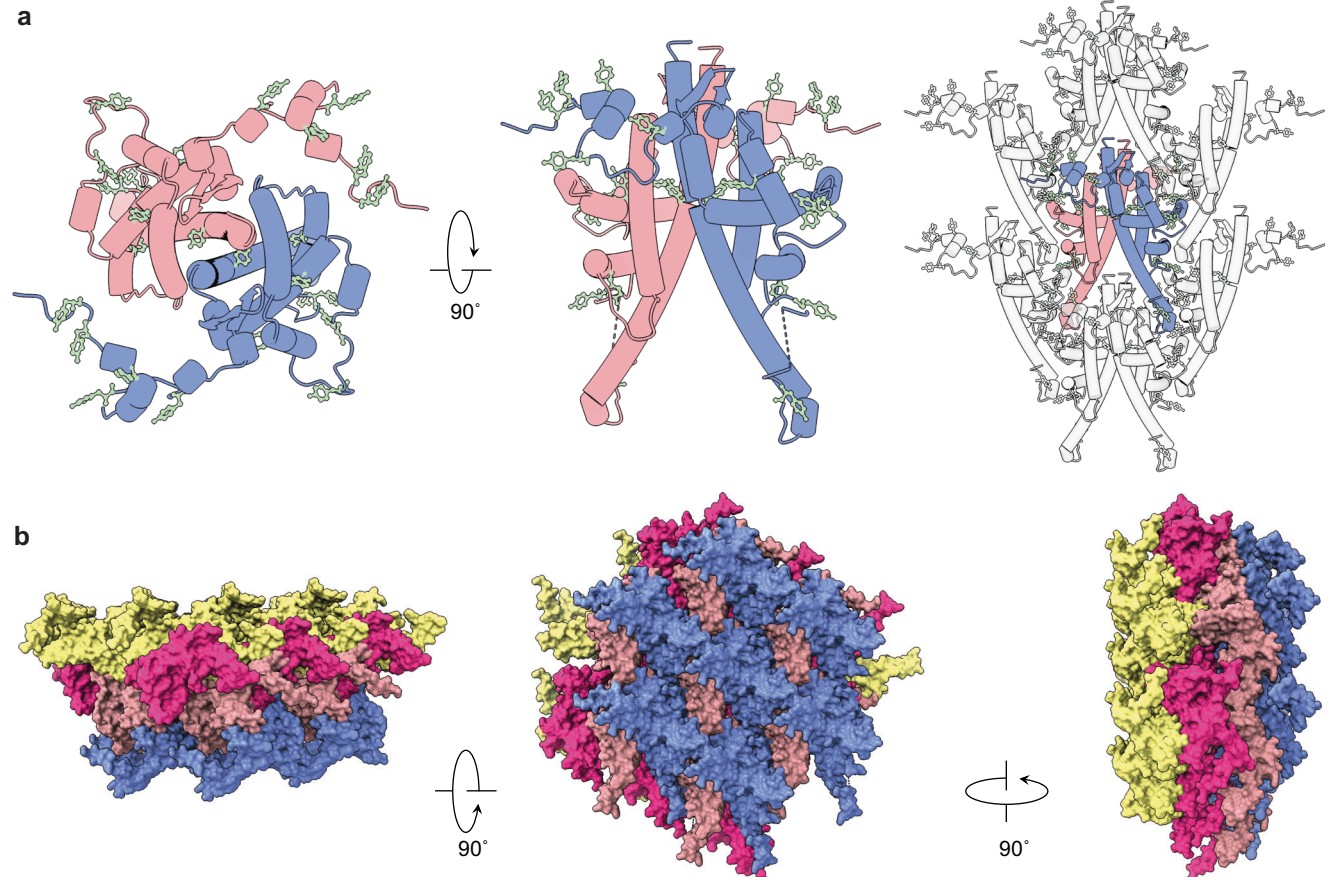

**Fig. 4 | The occlusion body is impermeable and poised for dissolution. a** Two polyhedrin molecules (red/blue) which form the dimeric repeating building block and position of tyrosine residues (pale green). Arrangement of the residues on the dimer and their resulting position in the lattice are shown. **b** Two sheets of the lattice (red/blue and pink/yellow) are shown from three orientations demonstrating that the assembled crystal does not contain any open channels.

sensitivity to high pH and $NaH_2CO_3$ is not a conserved property among nudivirus polyhedra, where polyhedra formed in shrimp are instead sensitive to acidic conditions[13]. It is unclear if this difference is due to the environmental stressors from a fully aquatic environment of the host organism or an effect of another environmental condition.

We next assessed the stability of the recombinant crystals under reducing and non-reducing conditions at high and neutral pH by SDS-PAGE (Fig. 3c). Under non-reducing conditions at neutral or high pH little material entered the gel. At neutral pH in the presence of reducing agent much of the sample ran as monomers or as small oligomers. With both reducing agent and at high pH over 90% of the protein ran as a monomer.

Previous studies have suggested tyrosine residues as the possible sensor of the basic pH in the midgut[7,8,31]. This is possible as tyrosine has a $pK_a$ of 10.1, so changes protonation state when the pH is increased. The increase in negative charge breaks hydrogen bonds and destabilises the interface. Analysis of our lattice shows many tyrosine residues (6% of the protein), with most clustering at protomer interfaces (Fig. 4a). Three tyrosine residues are found in the extended N-terminal region, which interacts with protomer II. A further two tyrosine residues are found in helix-$\alpha_2$ and helix-$\alpha_3$, which participate in domain switching. Finally, eight other tyrosine residues are observed across the various protein interfaces. A change in the protonation state of these residues would likely effect the packing and stability of the lattice.

During initial analysis of individual crystals of recombinantly expressed polyhedrin, we noted the presence of a crystal population (approximately 10% of crystals which could be indexed) where the unit cell had increased by 3.5 Å in two directions, giving cell dimensions of a = b = 57.2 Å, c = 105.8 Å. These crystals diffracted weakly, yielding a dataset to approximately 2.8 Å. Analysis of the structure showed significant portions of the structure were disordered. Residues 143–191 could not be resolved in the electron density maps, with helix-$\alpha_9$ and the N-terminal region of helix-$\alpha_{10}$ disordered (Supplementary Fig. 3c, d). The region between helix-$\alpha_5$ and helix-$\alpha_7$ was remodelled and helix-$\alpha_6$ was poorly resolved. At the N-terminus, only helix-$\alpha_2$ remained well resolved while the connecting loops and helix-$\alpha_1$ were not visible. The crystallographic disordering of these polyhedrin residues creates channels within the protein crystals. This increase in unit cell and the disordering of residues appears similar to that observed in a CPV4 polyhedra structure where similar changes occurred[9]. Whether this crystal form represents an assembly or disassembly intermediate, or an artefact of purification is unclear.

### Comparison to baculovirus and cypovirus polyhedra

Prior to this study, the structure of viral polyhedra had been determined for baculovirus and cypoviruses[7–9,12,32–37]. These both share a cubic symmetry (space group I23) with similar unit cell dimensions in the range 101–106 Å. The ToNV polyhedra in comparison form in the trigonal space group P3$_2$21 with unit cell dimensions a = b = 53.7 Å and c = 105.6 Å. An alternate indexing of the crystal with a pseudo-cubic (rectangular) lattice with cell dimensions a = 93 Å, b = 107.4 Å and c = 105.6 Å would give a cell with similar dimensions to the baculovirus and cypovirus (Supplementary Fig. 4). While it may be coincidence that there is an alternate indexing which gives an approximately cubic lattice with similar unit cell dimensions to previously determined OB, it is interesting to speculate that this may have been driven by evolutionary pressures.

The cypovirus and baculovirus lattices can be considered to utilise a repeating unit of a trimer to build their cubic lattice while ToNV contains a dimeric repeating unit (Supplementary Fig. 5a–c). The secondary and tertiary structures of baculovirus and cypovirus are predominantly β-strands which form the core of the protein, onto which long partially helical N-terminal extensions are appended. As discussed above, the ToNV polyhedrin structure is almost completely

helical, with a 3D fold not previously observed (to the best of our knowledge).

Though the differences are many, there are commonalities between the polyhedra. They all use a mixture of hydrophobic and electrostatic interactions to maintain the lattice. Domain swapping of the N-terminal extensions is observed in both the nudivirus and baculovirus polyhedra[8,12]. Symmetric and asymmetric disulfide bonds are observed in specific examples from the three virus families. Common to all polyhedra is the abundance of tyrosine residues at inter-subunit interfaces, presumably to destabilise the lattice under alkali conditions during the viral lifecycle[1]. Regarding the ToNV polyhedra structure, we observe that most of the tyrosine residues are at interfaces formed between the repeating dimeric unit and not within the dimer (Supplementary Fig. 3a).

Initial attempts to phase with models generated in silico were unsuccessful, therefore this sequence was submitted to the Critical Assessment of Protein Structure Prediction 15 (CASP15) event. ToNV ORF_059 was submitted as Target-1122 and 93 groups submitted models (Supplementary Fig. 1b). Visual assessment of the models showed that a small number of submitters were able to correctly predict the approximate secondary structure boundaries and relative locations. Parts of the polyhedrin core including parts of helix-$\alpha_5$, helix-$\alpha_8$, and helix-$\alpha_{10}$ were correctly positioned with equivalent Cα-RMSD positions of less the 2 Å (Supplementary Fig. 1c). Regions outside of these helices were often far away from their experimentally validated positions. None of the models from CASP15 or from Alphafold2 implementations (CollabFold v1.5.2) were able to produce a correct molecular replacement solution.

## Discussion

In this work, we have observed a different system of packing protein to form dense viral polyhedra and demonstrated that ToNV forms crystalline OBs. Comparison of the polyhedrin and polyhedra to those previously determined shows no structural conservation. It had been suggested from previous work that OB with cubic symmetry and a unit cell of 101–106 Å formed from a predominantly β-strand polyhedrin which conferred isotropic stability may have been a feature of a OB resulting from convergent evolution where environmental pressures selected for these properties[8,12]. With the inclusion of the ToNV polyhedra structure we show no conservation of protein secondary or tertiary structure, space group symmetry, and possibly unit cell dimension is required for polyhedra formation.

There are two apparent requirements of the polyhedrin monomer for correct formation of the OBs (i) to form a highly robust protein assembly and, (ii) to allow for the incorporation of elongated asymmetric rod-shaped virions in seemingly random orientations. The ToNV polyhedrin achieves the first requirement by forming a dense lattice, free from continuous channels, maintained in all directions by disulfide bonds, hydrophobic and electrostatic interactions, and domain swapping. Previously, Ji and collaborators suggested that an ideal protein size to allow both virus incorporation and dense lattice would be no larger than 40 kDa, and our study (ToNV polyhedrin is ~27.5 kDa) adds further weight to this statement[8].

To achieve the second requirement, OBs use a highly symmetric lattice. Thus, when the lattice is blocked from forming in some directions by the presence of a virion, the lattice can continue in many other directions. To allow integration of large asymmetric virions, a highly symmetric lattice, which will be maintained even when comparatively massive virions are present, is preferred both for ToNV in space group P3$_2$21 and for other viruses in space group I23. The embedded cypoviruses are icosahedral and approximately 60 nm in size[11], while baculoviruses and nudiviruses are asymmetric with dimensions of ~30–70 nm in width and 200-400 nm in length[11,16]. It is not clear why nudiviruses, but not the baculoviruses, evolved this very different polyhedra structure when the polyhedra needs to encapsulate

similarly rod-shaped virions. It may be that proteins other than the viral polyhedrin cause the observed difference in OB between nudivirus and baculovirus[17]. Future studies are required to understand the packaging signal or mechanisms used to encapsulate nudivirus virions.

## Methods

### Protein cloning and purification

A gene encoding the *Tipula oleracea* nudivirus ORF_059 (NCBI Reference Sequence: YP_009116706.1), that was previously identified as a functional homologue of the baculovirus polyhedrin and the major ToNV OB component[14,17], was synthesised (GeneArt) and cloned into the Multibac system to generate a baculovirus encoding the viral polyhedrin protein. For protein expression, Sf9 cells were grown in Sf-900 II serum-free media (Gibco), infected, and harvested after 72 h. To purify the crystals, the cell pellet was resuspended in water on ice for 20 min before being briefly sonicated prior to centrifugation. The pellet was then resuspended in a 0.1% (w/v) SDS at room temperature for 30 min. This process was repeated at least three times until the crystals were pure as assessed by inspection under a light microscope.

For production of selenomethionine derivatised protein, three mutations (F104M/L105M/L137M) were first introduced into the wild-type plasmid using standard cloning techniques. To produce seleno-methionine crystals, Sf9 cells were first infected with the mutant baculovirus for 18 h. The cells were then pelleted and resuspended in Sf-900 II serum-free media lacking methionine or cysteine (Gibco). After 4 h the cells were supplemented with cysteine and seleno-methionine to a final concentration of 150 mg/L and 500 mg/L, respectively. The crystals were purified as above for the wild-type samples.

The ToNV archival sample 35 was supplied as a crystal solution prepared by Kenneth M. Smith in the 1950s[38,39]. The sample was previously stored at the National Environmental Research Council, Centre for Ecology and Hydrology. Additional purification was not required.

### VMXm sample preparation

Native and selenomethionine protein crystal datasets were collected on the VMXm beamline. The crystal solutions were first diluted with ethylene glycol to a final concentration of 50% (v/v). Grids were prepared in a GP2 (Leica-microsystems) at 20 °C and 90% humidity. All samples were applied to freshly glow discharged R2/2 Cu 200 (Quantifoil) mesh grids. To prepare the sample, 2 μL of crystal solution was applied to the carbon side of the grids and 2 μL 50% (v/v) ethylene glycol applied to the back side. The grids were then blotted from the back for 7 s before being plunged into liquid ethane[40]. The grids were stored in liquid nitrogen until transfer into bespoke VMXm sample holders. Representative grids were imaged in a JEOL JSM-IT100 scanning electron microscope equipped with cryogenic sample stage and airlock (Quorum Technology). Uncropped and unedited TEM images are provided in the source data file.

### I24 sample preparation and data collection

The I24 beamline was used both to test lattice stability and to collect data on the two different unit cells observed in the wild-type recombinant crystals. The samples were prepared by pipetting ~100 nL of crystal slurry onto micromesh loops (MiTeGen) and wicking away excess liquid. The samples were flash cooled by plunging into liquid nitrogen. The crystal positions were identified by running grid scans. Data were collected at 1.5498 Å in order to try to use the sulfur anomalous signal to solve the structure. A total of 289 wedges of data were collected, consisting of 16° sweeps, 0.1° oscillations with 0.02 s exposure and a beam size of 7 mm × 7 mm. The data were collected on a DECTRIS PILATUS3 6 M detector and processed with *DIALS*[41] and *xia2.multiplex*[42].

### Protein structure prediction

Protein structures were predicted with trRosetta v1.0[24] or AlphaFold via ColabFold v1.5[25] using default settings and generating five models.

### Structure solution and determination

The SeMet data were collected at the VMXm beamline at Diamond Light Source. All datasets were collected on a DECTRIS EIGER2 X 9 M CdTe and processed using *DIALS* v1[41] and *xia2.multiplex* v1[42]. A single-wavelength anomalous diffraction experiment was carried out at 0.9787 Å where 40° sweeps from 55 selenomethionine derivatised crystals were used after initial triage of 78 crystals. Exposures of 0.02 s over a 0.1° oscillation were collected using a 3.5 μm × 3 μm beam. The structure was solved by CRANK v1.5[43] which found 3 of the 4 seleno-methionine sites that provided weak phases (FOM 0.158, which converged to 0.349 after density modification). These phases were good enough to automatically build a preliminary model (212 residues built with R/Rfree of 22.3/29.1). The structure was subsequently manually built in Coot[44] and refined in PHENIX[45]. Figures were prepared using PyMol 2.4.1[46] and ChimeraX v1.4-1.6[47].

Native data were collected on 65 of the 70-year-old crystals. The final dataset was composed of 16 data wedges, each with 200 images collected at 0.6328 Å wavelength using a 3.5 μm × 3 μm sized beam, with 0.2 s exposure and 0.1° oscillations. The SeMet protein model was used to phase the higher-resolution dataset.

### Viral polyhedra stability

To test the stability of the recombinant ToNV polyhedra, 20 μL of the crystal slurry was pelleted and the supernatant replaced with the test solution of interest or water (for freeze-thaw and 90 °C conditions). After incubation for the indicated amount of time, the solution was diluted with 50% (v/v) ethylene glycol before being applied to Micro-Meshes (MiTeGen) of various sizes. Excess liquid was removed with a wick prior to freezing in liquid nitrogen. Samples were sent to the I24 beamline at Diamond Light Source for diffraction experiments.

For the SDS-PAGE analysis, 15 μL of crystal slurry was mixed with 5 μL of either $NaH_2CO_3$ pH 10.5 or water and incubated at room temperature for 1 h. SDS sample loading buffer, either with or without 10 mM Dithiothreitol, was added to each sample. Samples were then boiled for 2 min before being applied to a NuPAGE 4–12% BisTris gel (ThermoFisher Scientific). Full uncropped SDS-PAGE images are provided in the Source data file.

### Reporting summary

Further information on research design is available in the Nature Portfolio Reporting Summary linked to this article.

## Data availability

All data are available from the corresponding authors and/or included in the paper or Supplementary information. Crystallographic data generated in this study have been deposited in the PDB including accession codes 8BBT (archival sample), 8BC5 (selenomethionine data), 8BCK (recombinant data), and 8BCL (recombinant data-expanded cell). Data generated in previous studies (PDB ID: 3JVB and PDB ID: 2OH5 are available in the PDB. Source data are provided with this paper.

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

## Acknowledgements

We thank Helen Ginn, Geoff Sutton, Ervin Fodor, David Stuart, and members of Grimes laboratory for helpful comments and discussion. This work was supported by Wellcome Investigator Award 200835/Z/16/Z (to J.M.G.). We thank Diamond Light Source for access to the MX beamlines (proposal numbers MX19946, NT23570, NT27314, MX28534).

## Author contributions

J.R.K., P.M., and J.M.G. designed the research. J.R.K., A.D.C., L.C., R.J.G., J.T., S.H., A.J.W., D.A., R.O., G.E., A.B., and J.M.G. performed the research, and all authors analysed the data. J.R.K. and J.M.G. wrote the paper with input from all authors.

## Competing interests

The authors declare no competing interests.
