## [Peer review file · Nature Communications]

REVIEWER COMMENTS

Reviewer #1 (Remarks to the Author):

The authors present the atomic structure of a nudivirus occlusion body protein determined from a 70-year-old crystal sample. The structure is novel as reflected by the attempts tried: unsuccessful molecular replacement by previously determined polyhedra structures, structures prediction by trRosetta and AlphaFold, and secondary structure fragments. It is hard to determine the structures of those less-studied virus proteins by both experiment and prediction. Thus this should be valuable work.

To be frank, I am not an expert in this field. Why the crystal is 70 years old? Does this mean the crystal was prepared 70 years ago? Who prepared the samples? These questions may be naïve. I can only comment on the aspect of structure prediction below.

This protein is a CASP15 target (ID: T1122), which is very challenging to predict for all participating groups. One of the reasons is the lack of any homologous sequences in the current sequence database. No reliable structure could be predicted even with trRosettaX-Single, the single-sequence version of trRosetta for orphan protein structure prediction (Wang et al, Nature Computational Science 2022). As we know, homologous sequences play a key role in structure prediction.

I am wondering if they could find some potentially homologous sequences of this protein using their expertise. If nothing could be found, it will be great if they could obtain some sequence relatives by sequencing. Conducting such an experiment may shed light on the structure determination for other proteins, given that sequencing is much cheaper than structure determination. However this is not mandatory.

Reviewer #2 (Remarks to the Author):

This paper presents a new structure of a novel type of virus polyhedron (a dense 3D lattice that serves to encapsulate and protect certain insect viruses). These are strange biological systems, and the addition of a new type is important in helping to delineate common vs variable architectural and symmetry features. The structural work appears to be sound. The use of microcrystal (VMX) diffraction methods adds interest. Overall this will add substantially to the structural biology literature. I have the following minor suggestions/corrections to consider:

1) On page 4 it is stated that there is one molecule *per unit cell*. This is incorrect. Presumably the authors mean one molecule *per asymmetric unit*.

2) The presumptive identification of a Ca²⁺ ion comes up on page 4. Please indicate something about the possible origin. Does it derive from crystallization conditions or do the authors assert that it represents a biological ion carried from cellular binding?

3) Page 5 discusses the arrangement of the crystal as a set of 2D layers, with successive layers related by the 3-fold screw axis of the P3(2)21 space group. It would add to the technical discussion to specify what the 2D layer group symmetry is of these layers. This reviewer believes the 2D layers exhibit c2 layer group symmetry. [Note the little c]

4) Consider abbreviating some of the highly detailed descriptions of structure around the bottom half of page 4 to the top part of page 5.

5) Line 10 on page 5 contains a non-sentence.

6) On page 7 where it is noted that predictions from CASP15 did not work in molecular replacement, it would be more impactful to say explicitly, “including AlphaFold”, since that is the leading tool; without indicating this a reader might not be able to discern whether the CASP entries included AlphaFold predictions. Also, state which version of AlphaFold is tested here. Newer versions show noted improvements, and a modern reader might wonder if the very latest version would produce a model that would work in MR; the authors could test this easily enough themselves (regardless of the CASP output).

7) In figure 2A,B, please show the underlying unit cell directions (labeled) and the 2-fold space group symmetry axes in order to help understand the packing symmetry.

8) Figure 4 refers to the molecular packing as “untraversable”. This is undefined. This wording should be removed unless a mathematically specific meaning can be provided and demonstrated to be the case here (e.g. by some kind of computation).

9) Please add a sentence somewhere to help explain to unfamiliar readers the essence of VMX data collection, i.e. that large numbers of microcrystals are held on a single grid (similar to those commonly

used for EM experiments) to enable rapid (narrow wedge) diffraction data collection in a fixed x-ray beam. Also, tie this in to Figure 1A.

Reviewer #3 (Remarks to the Author):

This manuscript by Keown and colleagues presents the structure of ToNV nudivirus polyhedra. The fold of this polyhedrin appears to be novel and the architecture of the virus-containing crystals is drastically different from the two known classes of viral polyhedra produced by cypoviruses and baculoviruses. The intricate sheet-like organisation is stabilised by “in-sheet” disulfide bond crosslinking and “inter-sheet” domain swapping. A particularly interesting point is highlighting the uniqueness of this structure is the (unusual) failure of modelling approaches by AlphaFold, trRosetta and CASP15 modellers.

I enjoyed reading this well-written manuscript. The findings establishes that at least three classes of viral polyhedra converged to crystals with similar functional features from unrelated proteins. It leaves open the conundrum of how this "orphan" protein evolved since there does not seem to be any other sequence related to the ToNV polyhedrin and no known homologous structure.

Along those lines, a couple of points deserve further analysis to support a key conclusion that this protein adopts a unique fold, which cannot be accurately modelled by current approaches:

1. The search for structural homologues is not described. While a reference to FoldSeek is present, the authors should detail what analyses were made (e.g. DALI, FoldSeek, PDBeFold; databases searched) and what criteria were used to conclude there are no homologous structures (e.g. Z or Q scores). Has any other database beyond the PDB been searched?

2. A prominent dimeric assembly is present in the crystal. It would be logical to include the modelling results for these, which may be more accurate.

I have the following additional comments:

The title focuses on the fact that the crystals are 70-year old.

This would only be remarkable if the crystals have been kept in conditions where “normal” protein crystals would decay. Have the crystals been kept frozen during this time? Freeze-thaw cycles are mentioned but no details are provided in the method section.

“Polyhedrin amino acid sequences in general have little sequence conservation and there is no evident homology between baculovirus and cypovirus polyhedrins [9].”

This was first shown in Ref 12. Reference 9 remained more ambiguous on this point noting the structural distance but building a phylogenetic tree including both baculovirus and cypovirus polyhedrins.

The result section starts without introducing the source of the material.

The authors should at least mention here where the crystals have been purified from, how they were stored (even if details are lacking given the 70-year period), their morphology/size and whether they contain virus particles as one would expect. The crystal preparation appears to be the same as the one described in Ref 14, which provides some nice SEM and TEM. If this is the case, imaging to confirm their morphology and the presence of virus particles is not necessary.

Similarly, the first sentence of this section mentions diffraction experiments without further details. While a structural biologist will guess what is meant from the rest of the paragraph, this should be clarified from the onset for readability (e.g. synchrotron X-ray diffraction rather than XFEL, microED etc.).

I am confused by the bottom panel of Fig. 2A. I understand it is a top view of the other panel but the cyan molecule is visible in the centre of the sheet.

The extra density away from the two “free” cysteines is not assigned (Fig. 2F). Is there any evidence for post-translational modifications from mass spectrometry data (e.g. from Ref17)? Is the density level consistent with a sulfur atom involved in a disulfide bond?

What is the point of Figure S4 and the associated text? Is it meant to suggest that the lattice originated as a cubic lattice and diverged to the current space group? Or, is it trying to point to a convergence towards (pseudo)-cubic symmetry? The discussion seems to conclude that the unit cell parameters and lattice symmetries are unrelated. This point needs to be clarified in the result section.

Stability experiment: it should be mentioned that this was tested on the recombinant crystals. Virus-containing crystals may have a different stability profile.

Reviewer #1 (Remarks to the Author):

The authors present the atomic structure of a nudivirus occlusion body protein determined from a 70-year-old crystal sample. The structure is novel as reflected by the attempts tried: unsuccessful molecular replacement by previously determined polyhedra structures, structures prediction by trRosetta and AlphaFold, and secondary structure fragments. It is hard to determine the structures of those less-studied virus proteins by both experiment and prediction. Thus this should be valuable work.

We thank the reviewer for their kind comments.

To be frank, I am not an expert in this field. Why the crystal is 70 years old? Does this mean the crystal was prepared 70 years ago? Who prepared the samples? These questions may be naïve. I can only comment on the aspect of structure prediction below.

These crystals were taken from a sample that was initial purified by Kenneth Smith in the 1950's. They were first previously characterised in *Bézier, A. et al. The genome of the nucleopolyhedrosis-causing virus from *Tipula oleracea* sheds new light on the Nudiviridae family. J Virol 89, 3008–3025 (2015)*. We have briefly described the provenance of the crystals as best we can in the methods section stating “The ToNV archival sample 35 was supplied as a crystal solution prepared by Kenneth M. Smith in the 1950s^{36,37}. The sample was previously stored at the National Environmental Research Council, Centre for Ecology and Hydrology.”

This protein is a CASP15 target (ID: T1122), which is very challenging to predict for all participating groups. One of the reasons is the lack of any homologous sequences in the current sequence database. No reliable structure could be predicted even with trRosettaX-Single, the single-sequence version of trRosetta for orphan protein structure prediction (Wang et al, Nature Computational Science 2022). As we know, homologous sequences play a key role in structure prediction.

I am wondering if they could find some potentially homologous sequences of this protein using their expertise. If nothing could be found, it will be great if they could obtain some sequence relatives by sequencing. Conducting such an experiment may shed light on the structure determination for other proteins, given that sequencing is much cheaper than structure determination. However this is not mandatory.

We, and our collaborators, have reached out to many nudivirus biologists however we have not been able to either find more samples to sequence or identify more sequences in existing databases. Our continuing work on this project is focused on identifying and determining more nudivirus OB structures to understand how we may generalise our results.

Reviewer #2 (Remarks to the Author):

This paper presents a new structure of a novel type of virus polyhedron (a dense 3D lattice that serves to encapsulate and protect certain insect viruses). These are strange biological systems, and the addition of a new type is important in helping to delineate common vs variable architectural and symmetry features. The structural work appears to be sound. The use of microcrystal (VMX) diffraction methods adds interest. Overall this will add substantially to the structural biology literature. I have the following minor suggestions/corrections to consider:

1) On page 4 it is stated that there is one molecule *per unit cell*. This is incorrect. Presumably the authors mean one molecule *per asymmetric unit*.

This is correct, we have changed this in the text.

2) The presumptive identification of a Ca²⁺ ion comes up on page 4. Please indicate something about the possible origin. Does it derive from crystallization conditions or do the authors assert that it represents a biological ion carried from cellular binding?

As these crystals are purified from native sources (protein crystallising inside the cell in which it's expressed) we presume this ion is picked up from the cell. With regards to the biological role of the calcium, we are unable to determine a possible role for this ion. We can say that the coordination does seem specific for a calcium ion. We note that calcium has been observed in OB crystals from other species (PDB ID 5A8V), where the biological importance was also not understood.

We have also corrected the stated coordination of the metal from octahedral to pentagonal bipyramidal as this was incorrect in the original text.

3) Page 5 discusses the arrangement of the crystal as a set of 2D layers, with successive layers related by the 3-fold screw axis of the P3(2)21 space group. It would add to the technical discussion to specify what the 2D layer group symmetry is of these layers. This reviewer believes the 2D layers exhibit c2 layer group symmetry. [Note the little c]

We have amended figure 3 to now show the corrected 3-fold symbol (triangle with extensions) and added arrows to indicate the direction of the 2-fold symmetry axis to both figure 2 and 3.

4) Consider abbreviating some of the highly detailed descriptions of structure around the bottom half of page 4 to the top part of page 5.

We thank the reviewer for this comment and appreciate the intention behind it. After much consideration we have decided to keep this detailed description. We believe the description and our figures, will aid readers in understanding the lattice when they are viewing the model.

5) Line 10 on page 5 contains a non-sentence.

We have rewritten the sentence into two sentences which now read “Each protomer forms four disulfide bonds, 34 hydrogen bonds, and 30 salt-bridges with adjacent protomers. These interactions bury approximately 48% or 7518 Å² of the surface area of the protomer.”

6) On page 7 where it is noted that predictions from CASP15 did not work in molecular replacement, it would be more impactful to say explicitly, “including AlphaFold”, since that is the leading tool; without indicating this a reader might not be able to discern whether the CASP entries included AlphaFold predictions. Also, state which version of AlphaFold is tested here. Newer version show noted improvements, and a modern reader might wonder if the very latest version would produce a model that would work in MR; the authors could test this easily enough themselves (regardless of the CASP output).

We have reworked this sentence to now read “None of the models from CASP15 or from Alphafold2 implementations (CollabFold v1.5.2) were able to produce a correct molecular replacement solution”.

7) In figure 2A,B, please show the underlying unit cell directions (labelled) and the 2-fold space group symmetry axes in order to help understand the packing symmetry.

We have added annotated unit cell directions to panels 2A, B to clearly define the unit cell direction. We have also added arrows and ellipses to panel A to better define the crystallographic 2fold and text to the caption to reflect these.

8) Figure 4 refers to the molecular packing as “untraversable”. This is undefined. This wording should be removed unless a mathematically specific meaning can be provided and demonstrated to be the case here (e.g. by some kind of computation).

We have rewritten the sentence to now read “The occlusion body is impermeable and poised for dissolution”.

9) Please add a sentence somewhere to help explain to unfamiliar readers the essence of VMX data collection, i.e. that large numbers of microcrystals are held on a single grid (similar to those commonly used for EM experiments) to enable rapid (narrow wedge) diffraction data collection in a fixed x-ray beam. Also, tie this in to Figure 1A.

We thank the reviewer for the comment and have added text stating “To prepare samples for the VMXm beamline, a slurry of microcrystals is applied to a single cryoEM grid that is then vitrified (Fig. 1A). The crystals on a single grid are then used for diffraction experiments, enabling the efficient collection of data from many protein crystals.”

Reviewer #3 (Remarks to the Author):

This manuscript by Keown and colleagues presents the structure of ToNV nudivirus polyhedra. The fold of this polyhedrin appears to be novel and the architecture of the virus-containing crystals is drastically different from the two known classes of viral polyhedra produced by cypoviruses and baculoviruses. The intricate sheet-like organisation is stabilised by “in-sheet” disulfide bond crosslinking and “inter-sheet” domain swapping. A particularly interesting point is highlighting the uniqueness of this structure is the (unusual) failure of modelling approaches by AlphaFold, trRosetta and CASP15 modellers.

I enjoyed reading this well-written manuscript. The findings establishes that at least three classes of viral polyhedra converged to crystals with similar functional features from unrelated proteins. It leaves open the conundrum of how this "orphan" protein evolved since there does not seem to be any other sequence related to the ToNV polyhedrin and no known homologous structure.

Along those lines, a couple of points deserve further analysis to support a key conclusion that this protein adopts a unique fold, which cannot be accurately modelled by current approaches:

1. The search for structural homologues is not described. While a reference to FoldSeek is present, the authors should detail what analyses were made (e.g. DALI, FoldSeek, PDBeFold; databases searched) and what criteria were used to conclude there are no homologous structures (e.g. Z or Q scores). Has any other database beyond the PDB been searched?

We used the three servers mentioned above including DALI, FoldSeek, and PDBeFOLD. In the manuscript we only mentioned FoldSeek as we believe this to the state of the art though tried all servers as there may be strengths and weaknesses to each search approach.

From Foldseek we searched the PDB100, AFDB50, AFDB-PROTEOME, AFDB-SWISSPROT, CATH50, and MGNIFY_ESM30 datasets using both 3Di/AA and TM-Align modes. From this search no hits with E-value <1, SeqID >20.6, or probability scores >0.6 were found. From DALI we searched the full PDB using default setting. The best Z-score was 5.2 with a Z-score of 3.3 over 112 residues. From PDBeFOLD the best hit had a Q-score of 0.049, Z-value of 1.6, over 62 residues.

We checked the top hits from each search program manually to confirm the scores were accurately representing the accuracy of the search. Most commonly the search had found a protein with an helical section which aligned to the C-terminal helix of the ToNV polyhedrin while connectivity of the other helices, location of helices running perpendicular to the C-terminal helix, and the N-terminal region were not accurately modelled.

We have added the following statement to the text and referenced the three servers used to make it clear what we tried “Secondary structure homology searches using the FoldSeek, Dali, or PDBeFold servers against the experimental structures in the Protein Data Bank or predicted models in the AlphaFold Protein Structure Database did not reveal any structures homologous to ToNV polyhedrin^{26–28}”

2. A prominent dimeric assembly is present in the crystal. It would be logical to include the modelling results for these, which may be more accurate.

During initial attempts to solve the data we were unsure which oligomer that would form the repeating unit and tried extensive trials of multimers ranging in size from dimer through hexamer. These were tried using Rosetta and AF2 algorithms optimised for both multimers and orphan sequences. None of these programs were able to provide a model which succeeded in molecular replacement. We have added a sentence “Oligomers of the ToNV polyhedrin sequence were trialled to generate additional *in silico* models but there were also unsuccessful in molecular replacement.” To make the readers aware these approaches were trialled and not successful.

I have the following additional comments:

The title focuses on the fact that the crystals are 70-year old.

This would only be remarkable if the crystals have been kept in conditions where “normal” protein crystals would decay. Have the crystals been kept frozen during this time? Freeze-thaw cycles are mentioned but no details are provided in the method section.

Establishing the exact provenance of the crystal sample has been quite difficult. We believe the appropriate references for the purification of this sample are Smith (SMITH KM. The structure of insect virus particles. *J Biophys Biochem Cytol.* 1956 May 25;2(3):301-6.) and Smith and Xeros (SMITH, K., XEROS, N. An Unusual Virus Disease of a Dipterous Larva. *Nature* **173**, 866–867 (1954)). These have been added to the manuscript.

With respect to storage conditions, we know the sample was stored at the NERC-CEH in the UK. We believe the sample was stored at room temperature from the 1950’s until it was shipped to France in the early 2010’s at which time it was kept frozen and likely underwent less than 5 freeze/thaw cycles.

“Polyhedrin amino acid sequences in general have little sequence conservation and there is no evident homology between baculovirus and cypovirus polyhedrins [9].”

This was first shown in Ref 12. Reference 9 remained more ambiguous on this point noting the structural distance but building a phylogenetic tree including both baculovirus and cypovirus polyhedrins.

The reviewer is correct and we have updated this reference used here.

The result section starts without introducing the source of the material.

The authors should at least mention here where the crystals have been purified from, how they were stored (even if details are lacking given the 70-year period), their morphology/size and whether they contain virus particles as one would expect. The crystal preparation appears to be the same as the one described in Ref 14, which provides some nice SEM and TEM. If this is the case, imaging to confirm their morphology and the presence of virus particles is not necessary.

As mentioned above, we have added two references to what we believe are the works describing the initial purification of the sample. "The ToNV archival sample 35 was supplied as a crystal solution prepared by Kenneth M. Smith in the 1950s^{36,37}. The sample was previously stored at the National Environmental Research Council, Centre for Ecology and Hydrology. Additional purification was not required."

These are the same crystals previously characterised in ref 14. We have added the sentence "Previous characterisation of the native OB showed particles with sizes of 2-5 microns with an irregular morphology that still contained infectious viral particles¹⁴" to briefly introduce the readers to the previous characterisation of this sample.

Similarly, the first sentence of this section mentions diffraction experiments without further details. While a structural biologist will guess what is meant from the rest of the paragraph, this should be clarified from the onset for readability (e.g. synchrotron X-ray diffraction rather than XFEL, microED etc.).

We have altered the starting sentence to now state "Initial X-ray experiments yielded high quality diffraction patterns..."

I am confused by the bottom panel of Fig. 2A. I understand it is a top view of the other panel but the cyan molecule is visible in the centre of the sheet.

The reviewer is correct this is a top view from the top panel of figure 2A. This was an issue of transparency for the molecules in the central column. We have regenerated the figure and we believe this problem has been corrected.

The extra density away from the two "free" cysteines is not assigned (Fig. 2F). Is there any evidence for post-translational modifications from mass spectrometry data (e.g. from Ref17)? Is the density level consistent with a sulfur atom involved in a disulfide bond?

We have no evidence for post-translational modification of these cysteines. The combination of extra density and the absence of the residues from the C-terminus we believe it more plausible for these to be disulfide bonds formed between the two C-terminal cysteines. The density is as would be expected for a covalent addition to the S atom.

What is the point of Figure S4 and the associated text? Is it meant to suggest that the lattice originated as a cubic lattice and diverged to the current space group? Or, is it trying to point to a convergence towards (pseudo)-cubic symmetry? The discussion seems to conclude that the unit cell parameters and lattice symmetries are unrelated. This point needs to be clarified in the result section.

The point we have tried to make is the second, that the OB lattices have converged to a pseudo-cubic symmetry. It is also correct that the crystallographic unit cells and lattice symmetries are unrelated between the species. We hypothesise that, perhaps for mechanical reasons, a discernible repeating unit of the polyhedra appear to be able to be reduced to an approximately cubic repeating unit.

We have added the text “ While it may be coincidence that there is an alternate indexing which gives an approximately cubic lattice with similar unit cell dimensions to previously determined OB, it is interesting to speculate that this may have been driven by evolutionary pressures.” To the section describing the alternate indexing of the lattice.

Stability experiment: it should be mentioned that this was tested on the recombinant crystals. Virus-containing crystals may have a different stability profile.

We have added the sentence “As these studies were conducted on the recombinant crystals and not the native OB we cannot exclude the possibility the native OB may behave differently.”

In addition, Table 1 is presently titled “Table 1. Stability of recombinant ToNV crystals” and in the text we refer to the experiment as “...assess the stability of the recombinant nudivirus polyhedra by exposing...”

REVIEWERS' COMMENTS

Reviewer #3 (Remarks to the Author):

I would have liked to see the alphafold model for the dimer (e.g. in Fig. S1) but it wasn't too long to generate myself. I agree it supports Fig. S1 and the overall conclusions.

The authors clarified all the points made in my initial review.

Reviewer #3 (Remarks to the Author):

I would have liked to see the alphafold model for the dimer (e.g. in Fig. S1) but it wasn't too long to generate myself. I agree it supports Fig. S1 and the overall conclusions.

The authors clarified all the points made in my initial review.

We thank the reviewer for their kind comments. We believe that given the low accuracy of the dimeric models, particularly the axis around which the dimer is generated, these do not enhance the clarity of the work.